# Boardgames as an innovative approach to promote life skills and well-being among inmates: A scoping review protocol

Carlo Andrea Pensavalle[1], Maria Giuliana Solinas[2]*, Christian Gardoni[3], Gabriele Giorgi[4], Tiziano Antognozzi[5], Federico Alessio[6]

1 Department of Chemistry, Physics, Mathematics and Natural Sciences, University of Sassari, Sardinia, Italy, 2 Department of Biomedical Sciences, University of Sassari, Sassari, Italy, 3 Department of Primary Education, European University of Rome, Rome, Italy, 4 Department of Human Sciences, European University of Rome, Rome, Italy, 5 AXES–Laboratory for the Analysis of Complex Economic Systems, IMT School for Advanced Studies Lucca, Lucca, Italy, 6 Business@Health Laboratory, European University of Rome, Rome, Italy

☉ These authors contributed equally to this work.
* mgsolinas1@uniss.it

**Data Availability Statement:** No datasets were generated or analysed during the current study, as we are presently in the process of publishing a protocol.

## Abstract

Over the past few decades, a growing body of evidence has emerged regarding the positive impact of boardgames in promoting life skills and well-being in various settings including health, education, and military schooling. However, the use of boardgames with inmates for cultural and educational purposes is still an unexplored and fragmented area of research. Incorporating boardgames into correctional settings can be a complex challenge for correctional stakeholders who seek to identify innovative tools to enhance inmate education and promote integration into society after incarceration. This article outlines our scoping review protocol designed to map and evaluate published and grey literature on the motivational, psychological, and pedagogical considerations involved in the design and use of boardgames as an innovative approach to promoting life skills and well-being among inmates.

## Introduction

The terms "Boardgames" or "Table-top Games" currently denote a well-established tradition of games characterized by a defined set of rules and physical components crafted for a specific play experience.

Given the prevailing popularity in the contemporary discourse of the term "digital games", boardgames are frequently described as "analog games", which means: "those game products that are not usually mediated through computer technologies, but which nevertheless exemplify contemporary cultural forms" [1]. Within this conceptual framework, empirical evidence related to boardgames can be situated alongside other diverse forms of games, such as role-playing games, trading card games, pervasive games, and similar entities.

While the origins of proto-boardgames and game components, such as dice, can be traced back to pre-language civilizations in the Eurasian Middle East and Ancient Egypt [2], it was

**Funding:** This work was supported by "Fondo di Ateneo per la Ricerca, 2020", University of Sassari, Italy. The funders did not and will not have a role in study design, data collection and analysis, decision to publish, or preparation of the manuscript. There was no additional external funding received for this study.

**Competing interests:** The authors have declared that no competing interests exist.

only in the early 20th century, with the advent of modern industrial production, that board-games began to achieve widespread commercial availability and recognition as distinct design products. BoardGameGeeks.com presently enumerates over 100,000 distinct board games available today, with descriptions and commentary on up to 130 game mechanics found within known boardgames [3]. In their examination of the relationship between game mechanics and specific forms of play, Samarasinghe et al. [4] emphasized that the contextual dynamics in which the game experience occurs also play a crucial role in determining specific impacts, such as well-being, education, and pro-sociality.

The social dynamics that shape our societies are intricately tied to the inclination of their members toward either prosocial or antisocial behaviors [5,6]. This complex interplay is inter-woven with the creative tendencies, experiences, and capabilities of individuals across diverse cultural, economic, and social dimensions [7,8]. While a significant body of experimental liter-ature on games has predominantly focused on digital games and their negative consequences (see Anderson et al., [9], for an extensive meta-analysis), recent investigations highlight the manifold impact of digital games on both antisocial and prosocial behavior, contingent upon contextual nuances [10,11].

Comparable to the predominant focus on digital games, a longstanding tradition among scholars has underscored empirical evidence illuminating the social advantages of analog games, reaching back to the era preceding digital games [12,13]. Booth's 2019 survey [14], exploring the motivations of boardgame players, emphasizes the substantial role of the social-izing aspect as a compelling motivator propelling engagement in gameplay. Specific choices in game design are proposed as potential correlatives [15]. The accumulating body of evidence indicates positive behavioral outcomes associated with boardgame activities in specific con-texts and social groups [16].

Kirriemuir and McFarlane's 2004 literature review [17] reports that through engagement in boardgame activities we can facilitate learning, providing a stimulating experience that capital-izes on the pleasure centers of our brains, enhancing learning abilities. Boardgames are regarded as effective learning tools, encouraging motivation, engagement, and behavior change while fostering understanding of complex systems [18–20]. Unlike digital games, boardgames promote face-to-face interactions with peers, tutors, or therapists, creating oppor-tunities for social interactions and life skills development [21,22]. Recent interest in board-game design as a problem-solving teaching tool has generated a variety of game design elements, aiming at creating engaging choices for players, and maintaining their involvement in the game activity, such as storytelling, mechanics, and player agency [23–27]. Through debriefing and follow-up discussion, boardgame sessions have the potential to facilitate knowl-edge transfer and reinforcement. Thereby, promoting well-being and life skills crucial for social reintegration into the complex and ever-changing modern society [28]. Boardgame learning approaches also cultivate creativity and empathy, essential characteristics for navigat-ing intricate systems of relationships and influencing each other's lives [29–31]. Existing litera-ture on boardgames in therapeutic contexts is extensive [32–44]. They have been employed in therapeutic settings for decades, serving either as a supplementary tool or as a central compo-nent also in the counseling process [45–47]. Further research is imperative to assess the effec-tiveness of boardgames as innovative interventions with positive outcomes in diverse populations, such as inmates. Individuals in confinement, or those with restricted freedom, encounter a multitude of mental and physical health challenges, often stemming from a lack of recreational or cultural activities as well as limited training opportunities [44]. Particularly for juvenile offenders, the learning environment must diverge from conventional schooling, and individuals with special education needs confront additional obstacles in the development of life skills [48–51].

Considering the above-mentioned potentiality of boardgames in generating positive effects on human behaviors, with this manuscript we intend to outline a scoping review protocol for mapping the literature on boardgames as tools for promoting life skills and well-being among inmates. It aims to investigate the motivational, psychological, and pedagogical considerations involved in the design and use of boardgames in incarcerated populations.

# Materials and methods

## Protocol design

According to Arksey and O'Malley [52], our scoping review consists of six stages: (1) the definition of the research questions and the objectives; (2) the identification of studies; (3) the selection of studies; (4) the extraction of data; (5) the synthesis and dissemination of the results, and (6) the consultation.

## Stage one: The definition of the research questions and the objectives

The main research objective is to identify, map, and synthesize the existing empirical literature on the design, use, and impact of boardgames to enhance life skills and well-being among inmates. To achieve this purpose, the following research questions will guide the study:

1. What empirical evidence is there on the design and impact of boardgames to improve life skills and well-being among inmates?

2. How are the different pedagogical, psychological, and technical factors addressed during the planning of boardgame activities for inmates?

3. What policy recommendations do derive from the collected evidence?

## Stage two: The identification of studies

**Eligibility criteria.**   As shown in Table 1, the review follows the Population Concept Context (PCC) framework to delineate the inclusion criteria, and to frame the review questions [53]. The primary search will be conducted on selected electronic databases. Products will be included in the review if they

**Table 1. The PCC framework.** The Population (or Participants)/Concept/Context (PCC) framework is recommended by the Joanna Briggs Institute (JBI) to identify the main concepts in primary review questions. The framework gives information about search strategies.

| Form field | Description |
|---|---|
| Review questions: | 1.What empirical evidence is there on the design and impact of boardgames to |
| Population (P): | improve life skills and well-being among inmates? |
| Concepts (C): | 2. How are the different pedagogical, psychological, and technical factors addressed |
| Context (C): | during the planning of boardgame activities for inmates? |
| Languages: | 3. What policy recommendations do derive from the collected evidence? |
| Date of publication: | Inmates |
| Type of publication or | Use of boardgames to develop life skills and well-being among inmates. |
| source: | Penitentiary institutions |
| Domains addressed/focus of | Italian, English, Spanish |
| study: | January 2012-December 2023 |
| What result: | Research articles, evaluation reports, project reports, government reports, book chapters, and conference articles. |
|  | Boardgames, life skills, well-being |
|  | Key findings that relate to the scoping review questions. |

a.  are published starting from year 2012.

b.  are published in Italian, English, or Spanish.

c.  address boardgames as life skills and well-being tools used among inmates.

d.  report qualitative and quantitative outcomes and provide an analysis of empirical data on inmates' life skills and well-being.

The type of publications that will be included in the review are research articles, evaluation reports, project reports, government reports, book chapters, and conference articles.

Qualitative studies will be also included, and their reference lists will be screened for potential eligible studies.

**Information sources.**   A three-stage process will be used to search and identify potentially relevant studies. The University librarians will be consulted regarding the following electronic databases: PsycINFO, ERIC (Education Resources Information Center), SciELO, and Education Source, via Ebscohost simultaneously.

Separate searches will also be conducted via JSTOR, Emerald, Science Direct, DOAJ, IEEE Xplore, OECD Library, Springer Link, Taylor & Francis, ACM Digital Library. In addition, searches via Medline, Scopus, and Web of Science will be run to cover additional journals not included within the above database.

The first stage will cover the search of qualitative and quantitative studies published in Italian, English, or Spanish, since January 1$^{st}$, 2012. Qualitative studies could be an important source of information too, about barriers to the use and design of boardgames, or attitudes, and beliefs about the use of these tools with inmates, all of which could play an important role in the decision-making process.

The second stage will consist of citation mining of documents identified during the primary stage. This includes a manual backward and forward search of references cited in both previous reviews and in studies selected for this review.

The third stage will focus on the identification of grey literature and will be conducted in ProQuest, Google Scholar, and Semantic Scholar.

The search will use the inclusion criteria described before. Two reviewers will independently carry out this process and, if any disagreement arises, it will be resolved via mutual discussion.

## Stage three: The selection of studies

The selection of studies will proceed accordingly to PRISMA-ScR guidelines [54] as shown in the following flow-diagram (Fig 1). In the identification stage, the title will be screened for topic relevance. In the screening stage, the abstract will be read only if the title is in line with the objectives of the review. In the eligibility stage, the reviewers, referring to the PCC framework inclusion criteria, will independently read the full text of the studies selected, identifying the records included in the review.

## Stage four: The extraction of data

The extracted data will be classified as shown in Fig 2, under the main categories of the Yusoff, et al. conceptual framework [55], adapted to boardgames, to better understand the different approaches used to design, develop, and implement boardgames, and to measure their impact on life skills improvements. This process will be carried out in duplicate by independent reviewers. Any discrepancies will be discussed until agreement will be reached.

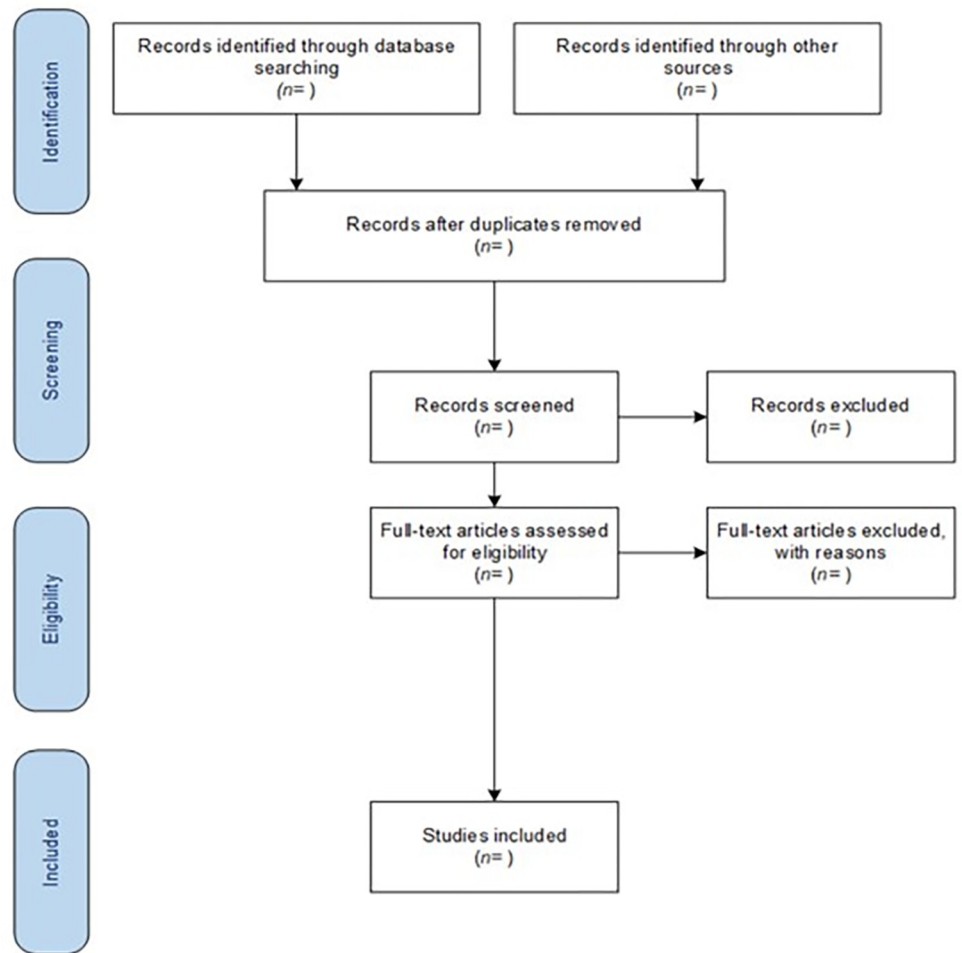

**Fig 1. PRISMA Flow Diagram for Scoping Review process.** The PRISMA Flow Diagram for the Scoping Review process is a flowchart used to document the sources used in scoping reviews, and the selection and screening processes.

## Stage five: The synthesis, and dissemination of results

In this stage, the inclusion/exclusion pathway will be presented using the flow diagram in Fig 1. An overview table will be created to display information about study characteristics from retrieved documents. This table will display information about location, author, year of publication, title, and sources [56]. Reviewers will discuss and consolidate the results. We will employ a narrative strategy to summarize, and to synthetize the data about the pedagogical, psychological, and technical approaches reported in the documents included.

## Stage six: The consultation

According to the Joanna Briggs Institute [57], this step involves a knowledge translation activity, which is a model of collaborative research, where researchers work with knowledge users who identify a problem and have the authority to implement the research recommendations. Therefore, we will collaborate closely with a reference group consisting of relevant stakeholders in the field of Correctional Systems. We will communicate with this group in three main phases: (1) in the search process to obtain input on relevant keywords and grey literature; (2) in the investigation process to ensure knowledge translation and (3) in the end process to

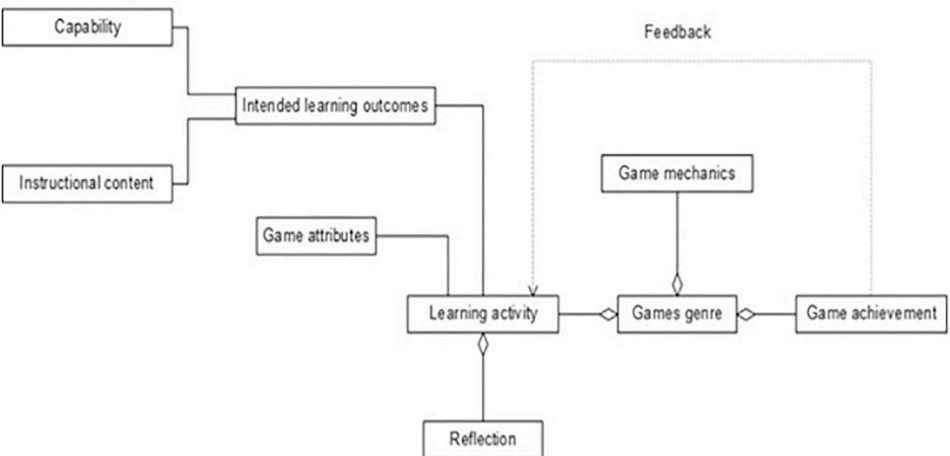

**Fig 2. Yusoff Conceptual framework.** The Yusoff Conceptual framework, allows to clearly describe connections among the ideas about the effectiveness of boardgames as innovative interventions tools with positive outcomes and makes apparent the significance of the study.

inform and discuss interpretation of the findings. The reference group has representatives from different organizations: detention facilities (prison directors, medical staff, and educators), universities (university penitentiary delegates), and voluntary social associations (prison volunteers).

**Quality assessment.** Quality assessment of the included articles will not be assessed, as it is outside the scope of this scoping review.

## Limitations and implications

There are few limitations that should be mentioned. The study will not take into consideration categorical differentiations on the nature of boardgame activities, (i.e., their cooperative or competitive nature). As it could be hypothesized that different categories of boardgames could have different effects and effectiveness [58]. Future developments could also consider these differences as variables of interest.

Also, the authors are aware of potential limitations due to the heterogeneity of approaches and population's types of inmates [59]. These factors can all influence the methods applied, the design and development of boardgame activities and the measures to assess their impact on life skills improvement and well-being among inmates.

## Discussion

This protocol outlines a clear method for examining the impact of boardgames on life skills and well-being among inmates. The results following this study could guide the direction of future research and furnish educational professionals, game designers, and decision-makers with evidence-based insights to make choices that point out on the opportunity of improving inmate's communication skills, conflict resolution strategies, emotional intelligence, time management, team building, and motivation through boardgames.

## Supporting information

**S1 Checklist. PRISMA-P (Preferred Reporting Items for Systematic review and Meta-Analysis Protocols) 2015 checklist: Recommended items to address in a systematic review**

**protocol\*.**
(PDF)

**S2 Checklist. Preferred Reporting Items for Systematic reviews and Meta-Analyses extension for Scoping Reviews (PRISMA-ScR) checklist.**
(PDF)

## Author Contributions

**Writing – review & editing:** Carlo Andrea Pensavalle, Maria Giuliana Solinas, Christian Gardoni, Gabriele Giorgi, Tiziano Antognozzi, Federico Alessio.

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
