## [Decision Letter · Decision Letter 0]

5 Dec 2023

PONE-D-23-26833Boardgames as an innovative approach to promote life skills and well-being among inmates: A scoping review protocolPLOS ONE

Dear Dr. Solinas,

Thank you for submitting your manuscript to PLOS ONE. After careful consideration, we feel that it has merit but does not fully meet PLOS ONE’s publication criteria as it currently stands. Therefore, we invite you to submit a revised version of the manuscript that addresses the points raised during the review process.

The article is well structured, the methodology applied is correct, and the topic covered is very relevant. Only minor changes are required.

We look forward to receiving your revised manuscript.

Kind regards,

Andrea Cioffi

Academic Editor

PLOS ONE

Journal Requirements:

"This work was supported by “Fondo di Ateneo per la Ricerca, 2020”, University of Sassari, Italy. "

Reviewers' comments:

Reviewer's Responses to Questions

**Comments to the Author**

1. Does the manuscript provide a valid rationale for the proposed study, with clearly identified and justified research questions?

Reviewer #1: Yes

Reviewer #2: Yes

2. Is the protocol technically sound and planned in a manner that will lead to a meaningful outcome and allow testing the stated hypotheses?

Reviewer #1: Yes

Reviewer #2: Yes

3. Is the methodology feasible and described in sufficient detail to allow the work to be replicable?

Reviewer #1: Yes

Reviewer #2: Yes

4. Have the authors described where all data underlying the findings will be made available when the study is complete?

Reviewer #1: Yes

Reviewer #2: Yes

5. Is the manuscript presented in an intelligible fashion and written in standard English?

Reviewer #1: Yes

Reviewer #2: Yes

6. Review Comments to the Author

You may also provide optional suggestions and comments to authors that they might find helpful in planning their study.

Reviewer #1: Introduction

Please give examples of boardgames and it will be nice to mention the ones intended for this scoping work. Board games as a word is too broad.

Line 41-extensive (where?)

Line 42 -you mention other populations meaning they have been used in some populations-please mention them to offer context.

Generally, it is not very clear to me why the authors are interested in board games in prisons -please provide context of whether some formative work has been done and you want to prove hypothesis-there must always be reason for wanting to do something.

Methods

Well described and I like the fact they have suggested searching many different data bases, checking reference lists and using grey literature to increase the scope of the study.

References

The reference list is untidy, please tidy up.

Reviewer #2: You can use different keywords as the ones you used are already mentioned in the title.

Please check for small typos (for example line 41-thei_r, line 72-diff_erent, line 108-"gray"->"grey").

Also, it would be more scientific if you used some references when you mention a potential problem that can occur during the study (for example line 175->...effectiveness (REF)., line 180->...among inmates (REF).

In table 1, it would be useful to align the elements of the left column with those of the right.

Finally, I would suggest making the legends of your tables and figures a little more descriptive to make them clear to the reader.

7. PLOS authors have the option to publish the peer review history of their article (what does this mean?). If published, this will include your full peer review and any attached files.

Reviewer #1: **Yes: **Rachel Kawuma

Reviewer #2: No

---

## [Author Response · Author response to Decision Letter 0]

4 Jan 2024

Responses to Editor comments.

In the revised version of our paper, which we upload in the submission needing a revision folder, we followed the suggestions kindly provided by the reviewers. 

We specify that there was no additional external funding received for this study. This work was only supported by “Fondo di Ateneo per la Ricerca, 2020”, University of Sassari, Italy. 

We would like to clarify the following point 3 in your letter: “We note that you have stated that you will provide repository information for your data at acceptance. Should your manuscript be accepted for publication, we will hold it until you provide the relevant accession numbers or DOIs necessary to access your data. If you wish to make changes to your Data Availability statement, please describe these changes in your cover letter and we will update your Data Availability statement to reflect the information you provide.”, regarding the provision of repository information for our data upon acceptance. As we are presently in the process of publishing a protocol, the data collection phase has not yet begun. Following the completion of the scoping review, and in accordance with our intention to submit the results to your journal for future publication, we will promptly furnish all the necessary information to make the repository data accessible.

All authors have approved this revised version of our original manuscript and agree with its re-submission to the PLOS ONE journal, read and understood your journal’s policies. 

Response to Reviewer#1: 

Introduction

Please give examples of boardgames and it will be nice to mention the ones intended for this scoping work. Board games as a word is too broad.

R. We really thank the reviewer for showing interest in our research topic. We took inspiration from this observation to clarify the meaning of the term “boardgame” in the revised introduction of the re-submitted paper. 

In few Italian detention facilities, located in the north and central part of Italy (Firenze, Genova, Imperia, Milano, Ravenna, Saluzzo, Sanremo) there are a variety of modern boardgames played in ongoing experimental activities with inmates. Here there is a list of some of the most frequently used: Cartagena, Stone Age, Carcassonne, Catan, Saboteur, Whitehall Mystery, Camel Up, Splendor, The Island, Ticket to Ride: Europe,7 Wonders Architects, Crossing, Project L, Azul, Flamme Rouge, For Sale, The Great Dalmuti, Avalon, The Resistance, Celestia, Cerberus, Deep Sea Adventure, etc. 

On the BoardGameGeeks.com site is possible to find the technical details of each one of the boardgames listed above. 

However, as reported in the submitted paper, our goal is to outline a scoping review protocol for mapping the literature on boardgames as tools for promoting life skills and well-being among inmates, in order to share with other researchers a standard approach to investigate and collect information around the world about which boardgames are in use, how they are used, what empirical evidence is there on the design and impact of boardgames to improve life skills and well-being among inmates, how are the diﬀerent pedagogical, psychological, and technical factors addressed during the planning of boardgame activities for inmates, what policy recommendations do derive from the collected evidence.

We respectfully consider it necessary to remain 'open' to any kind of boardgames we might find in use to be included in this work, especially because it is a Scoping Review. It seems necessary to stay open to all types of boardgames we will find, especially because the literature in this area is limited. This is evident from the fact that detention systems are enclosed and complex environments with unique limitations.

Line 41-extensive (where?)

R. We thank the reviewer for the encouraging clarification requested. Existing literature on boardgames in therapeutic contexts is extensive. We added few words and new references to the one already mentioned in the introduction of the re-submitted paper. Boardgames have been employed also in counseling process for decades, serving either as a supplementary tool or as a central component (Swank, J. M., & Weaver, J. L. (2021). In H. G. Kaduson & C. E. Schaefer (Eds.), Play therapy with children: Modalities for change (pp. 209–223). American Psychological Association. https://doi.org/10.1037/0000217-014; Bratton SC, Ray D, Rhine T, Jones L. The efficacy of play therapy with children: A meta-analytic review of treatment outcomes. Prof Psychol Res Pract 2005; 36:376–390.14; Nickerson ET, O’Laughlin KB. It’s fun—But will it work?:The use of games as a therapeutic medium for children and adolescents. J Clin Child Psychol 1980; 9:78–81.15; Wilde J. The effects of the Let’s Get Rational board game on rational thinking, depression, and self-acceptance in adolescents. J Ration Cogn Ther 1994; 12:189–196); (as reported in the revised introduction of the re-submitted paper).

Line 42 -you mention other populations meaning they have been used in some populations-please mention them to offer context.

Generally, it is not very clear to me why the authors are interested in board games in prisons -please provide context of whether some formative work has been done and you want to prove hypothesis-there must always be reason for wanting to do something.

R. Thank you again for the opportunity to better explain why we are interested in boardgames in prison. Research in this field could provide valuable insights into the potential benefits of boardgames for incarcerated populations, including their impact on social skills, mental health, and overall rehabilitation. 

Examples of other populations include:

Juvenile Offenders: Research could explore how boardgames can be used as rehabilitative tools for juvenile offenders, fostering skill development, and reducing recidivism.

Adult Inmates in Correctional Facilities: Investigating the impact of boardgames on the social dynamics and mental well-being of adult inmates, promoting positive interactions and stress relief.

Inmates with Substance Abuse Issues: Exploring the potential of boardgames in addressing substance abuse issues among incarcerated individuals, complementing existing rehabilitation programs.

Women in Correctional Facilities: Studying how boardgames may contribute to the well-being and social integration of female inmates, considering gender-specific aspects of rehabilitation.

Inmates with Mental Health Conditions: Assessing the therapeutic benefits of boardgames for incarcerated individuals dealing with mental health challenges, potentially aiding in coping and rehabilitation.

Incarcerated Veterans: Researching the use of boardgames as a means of support and rehabilitation for incarcerated individuals who are military veterans, addressing their unique needs.

Long-Term Inmates: Examining the role of boardgames in maintaining mental stimulation and preventing social isolation among inmates serving long-term sentences.

Pre-Release Programs: Investigating the incorporation of boardgames into pre-release programs to prepare inmates for reintegration into society, focusing on social and cognitive skills.

Family Visitation Centers: Studying how boardgames can facilitate positive family interactions during visitation, contributing to the maintenance of family bonds despite incarceration.

Community Reentry Programs: Exploring the use of boardgames as part of community reentry programs, aiding in the transition from incarceration to society and promoting social integration.

The fundamental idea which drives our work is collecting concrete evidence regarding the potential benefits of boardgames in these specific populations. Through a scoping review that showcases existing literature, we intend to encourage governments, administrations, and any other entity/organization dealing with incarcerated individuals to consider of investing on a simple and cost-effective set of tools (boardgames). Through their ambivalence in effectiveness and benefits, these tools might contribute both to improving a higher level of 'humanity' in a rather critical context, to promoting well-being and to facilitating the development of life skills, aiming for an easier reintegration into social and work environments once the period of incarceration has concluded.

References

The reference list is untidy, please tidy up.

R. We thank the reviewer and we proceeded to tied it up with the integration of new updated references.

Response to Reviewer #2: 

You can use different keywords as the ones you used are already mentioned in the title

R. We thank the reviewer, and we made the suggested changes.

Please check for small typos (for example line 41-thei_r, line 72-diff_erent, line 108-"gray"->"grey").

R. We thank the reviewer, and we check for the typos.

Also, it would be more scientific if you used some references when you mention a potential problem that can occur during the study (for example line 175->...effectiveness (REF)., line 180->...among inmates (REF).

R. We thank the reviewer, and we add the references respectively into the text of the re-submitted paper.

In table 1, it would be useful to align the elements of the left column with those of the right.

R. We thank the reviewer, and we took care of that.

Finally, I would suggest making the legends of your tables and figures a little more descriptive to make them clear to the reader.

R. We thank the reviewer, and we took care of that too as follows:

Tab 1 The Population (or Participants)/Concept/Context (PCC) framework is recommended by the Joanna Briggs Institute (JBI) to identify the main concepts in primary review questions. The framework gives information about search strategy.

Fig 1: The PRISMA Flow Diagram for the Scoping Review process is a flowchart used to document the sources used in scoping reviews, and the selection and screening processes.

Fig 2: The Yusoﬀ, et al. Conceptual framework, which allows to clearly describe connections among the ideas about the effectiveness of boardgames as innovative interventions tools with positive outcomes and makes apparent the significance of the study.

---

## [Decision Letter · Decision Letter 1]

30 Jan 2024

Boardgames as an innovative approach to promote life skills and well-being among inmates: A scoping review protocol

PONE-D-23-26833R1

Dear Dr. Solinas,

We’re pleased to inform you that your manuscript has been judged scientifically suitable for publication and will be formally accepted for publication once it meets all outstanding technical requirements.

Kind regards,

Andrea Cioffi

Academic Editor

PLOS ONE

Additional Editor Comments (optional):

Reviewers' comments:

Reviewer's Responses to Questions

**Comments to the Author**

1. Does the manuscript provide a valid rationale for the proposed study, with clearly identified and justified research questions?

Reviewer #1: Yes

Reviewer #2: Yes

2. Is the protocol technically sound and planned in a manner that will lead to a meaningful outcome and allow testing the stated hypotheses?

Reviewer #1: Yes

Reviewer #2: Yes

3. Is the methodology feasible and described in sufficient detail to allow the work to be replicable?

Reviewer #1: Yes

Reviewer #2: Yes

4. Have the authors described where all data underlying the findings will be made available when the study is complete?

Reviewer #1: Yes

Reviewer #2: Yes

5. Is the manuscript presented in an intelligible fashion and written in standard English?

Reviewer #1: Yes

Reviewer #2: Yes

6. Review Comments to the Author

You may also provide optional suggestions and comments to authors that they might find helpful in planning their study.

Reviewer #1: I have found the response by authors sufficient and manuscript reads well after providing context why they were interested in this particular topic. this scoping review will be useful to start up such interventions among populations such as inmates who are incarcerated. They have also improved the reference list which strengthens their paper.

Reviewer #2: In my opinion this interesting research is now presented in an understandable manner. The use of english language is well enough and the methodology is well described.

7. PLOS authors have the option to publish the peer review history of their article (what does this mean?). If published, this will include your full peer review and any attached files.

Reviewer #1: No

Reviewer #2: No

---

## [Editor Report · Acceptance letter]

20 Feb 2024

PONE-D-23-26833R1 

PLOS ONE

Dear Dr. Solinas, 

I'm pleased to inform you that your manuscript has been deemed suitable for publication in PLOS ONE. Congratulations! Your manuscript is now being handed over to our production team.

Kind regards, 

on behalf of

Dr. Andrea Cioffi 

Academic Editor

PLOS ONE